

# Feral frogs, native newts, and chemical cues: identifying threats from and management opportunities for invasive African Clawed Frogs in Washington state

David Anderson[1,*], Olivia Cervantez[1,*], Gary M. Bucciarelli[2], Max R. Lambert[3] and Megan R. Friesen[1]

[1] Department of Biology, Saint Martin's University, Lacey, WA, USA
[2] Department of Wildlife, Fish, and Conservation Biology, University of California, Davis, CA, United States of America
[3] Science Division, Washington Department of Fish and Wildlife, Olympia, WA, USA
[*] These authors contributed equally to this work.

Corresponding author
Megan R. Friesen,
MFriesen@stmartin.edu

## ABSTRACT

Invasive species threaten biodiversity globally. Amphibians are one of the most threatened vertebrate taxa and are particularly sensitive to invasive species, including other amphibians. African clawed frogs (*Xenopus laevis*) are native to Southern Africa but have subsequently become invasive on multiple continents—including multiple parts of North America—due to releases from the pet and biomedical trades. Despite their prevalence as a global invader, the impact of *X. laevis* remains understudied. This includes the Pacific Northwest of the USA, which now hosts multiple expanding *X. laevis* populations. For many amphibians, chemical cues communicate important information, including the presence of predators. Here, we tested the role chemical cues may play in mediating interactions between feral *X. laevis* and native amphibians in the Pacific Northwest. We tested whether native red-legged frog (*Rana aurora*) tadpoles display an antipredator response to non-native frog (*X. laevis*) or native newt (rough-skinned newts, *Taricha granulosa*) predator chemical stimuli. We found that *R. aurora* tadpoles exhibited pronounced anti-predator responses when exposed to chemical cues from *T. granulosa* but did not display anti-predator response to invasive *X. laevis* chemical cues. We also began experimentally testing whether *T. granulosa*—which produce a powerful neurotoxin tetrodotoxin (TTX)—may elicit an anti-predator response in *X. laevis*, that could serve to deter co-occupation. However, our short-duration experiments found that *X. laevis* were attracted to newt chemical stimuli rather than deterred. Our findings show that *X. laevis* likely poses a threat to native amphibians, and that these native species may also be particularly vulnerable to this invasive predator, compared to native predators, because toxic native newts may not limit *X. laevis* invasions. Our research provides some of the first indications that native Pacific Northwest species may be threatened by feral *X. laevis* and provides a foundation for future experiments testing potential management techniques for *X. laevis*.

## INTRODUCTION

Invasive species threaten biodiversity globally (*Didham et al., 2005*; *Didham et al., 2007*; *Pyšek & Richardson, 2010*; *Ahmed et al., 2022*). While some effects of invasive species on native species and ecosystems are easily recognizable, other effects are challenging to identify. In some cases, native species responses that are behaviorally mediated may not be easily measured (*Simberloff, 2013*). Understanding the impacts of invasive species on a particular species or ecosystem is essential for appropriately allocating resources and coordinating management efforts (*Epanchin-Niell, Englin & Nalle, 2009*).

Amphibians globally have experienced tremendous losses and an estimated 41% of amphibian species are listed as threatened on the International Union for Conservation of Nature Red List (*IUCN, 2024*). Invasive species have contributed greatly to these declines as roughly 16% of threatened amphibian declines and approximately 30% of amphibian extinctions are at least partially attributed to invasive species (*Falaschi et al., 2020*). The threat of invasive species to amphibians may be greatest from aquatic invasive predators (*Kats & Ferrer, 2003*) due to predation, competition, hybridization, and disease (*Falaschi et al., 2020*). In North America, for example, native amphibians are not only threatened by invasive species in general but by competition with and predation by invasive amphibians. For example, bullfrogs (*Lithobates catesbeiana*) are invasive in the Pacific Northwest and threaten common amphibians, like red-legged frog (*Rana aurora*), to imperiled species like the Oregon spotted frogs (*Rana pretiosa*; *Meshaka et al., 2022*). The overall threat amphibians face globally and the impact of invasive species on this taxa has, in some circumstances, given rise to conservation interventions.

Management approaches for aquatic invaders have been trialed to control the impact on native species. A range of management techniques are used, including trapping and removal or euthanization, habitat modification, and chemical poisoning (*Adams & Pearl, 2007*; *Lorrain-Soligon et al., 2021*; *Ojala-Barbour et al., 2021*). One technique used with varying success for a range of invasive species includes the use of biocontrols. Biocontrols are living organisms that are introduced to an area or whose populations are enhanced to reduce an invasive species' population or impact (*Stoner, 2023*). While some biocontrol management plans have introduced new problems to ecosystems, the use of native biocontrols has been a successful approach in others (*Messing & Wright, 2006*). For example, large-bodied groupers (*Epinephelus striatus* and *Mycteroperca tigris*) have been found to actively consume invasive lionfish (*Pterios* spp.*) in the Caribbean (*Mumby, Harborne & Brumbaugh, 2011*). As such, the act of helping to amplifying native species in some locations may bolster biocontrol efforts.

Animal behavior analyses have become essential tools for conservation and have aided in identifying the impacts of invasive species and effective management techniques (*Holway & Suarez, 1999*; *Berger-Tal et al., 2011*). Amphibians are a model species for

understanding the role of chemical cues in mediating predator–prey relationships and various non-consumptive interactions (*Kiesecker, Chivers & Blaustein, 1996*; *Grayson et al., 2012*). For instance, when presented with a visual cue, western toad (*Anaxyrus boreas*) tadpoles did not exhibit antipredator behavior, however in the presence of a predator chemical cue they display avoidance behaviors (*Kiesecker, Chivers & Blaustein, 1996*). These same types of analyses can be informative for understanding invasive species impacts as well. For example, Pacific chorus frog (*Pseudacris regilla*) tadpoles exhibit avoidance behavior when exposed to chemical cues of invasive bullfrogs (*L. catesbeiana*; *Chivers et al., 2001*). Further, *R. aurora* tadpoles exhibited high anti-predator refuge use behavior in response to both native and invasive fish and crayfish predator chemical cues, whereas chorus frog (*P. regilla*) tadpoles only responded to native fish predators but not invasive fish or crayfish chemical cues (*Pearl et al., 2003*). *R. aurora* also showed an increase in antipredator behavior when introduced to chemical cues for metabolic waste of conspecific tadpoles, showing a reduction in movement as a main response (*Kiesecker et al., 1999*), yet their behavioral responses to introduced bullfrogs appeared to vary by population (*Kiesecker & Blaustein, 1997*). The ability to identify and apply anti-predator responses can provide native amphibian populations a critical survival advantage, however—and from the perspective of the invader—the ability for non-native populations of amphibian to be able to identify the chemical cues of native threats also would provide them a survival advantage. For example, studies have found that non-native amphibians, like *L. catesbeiana* and Coquí (*Eleutherodactylus coqui*), sometimes cannot recognize cues from native predators (*Garcia et al., 2012*; *Marchetti & Beard, 2021*). *L. catesbeiana* recognized fish predators (*e.g.*, largemouth bass, *Micropterus salmoides*) only if they were from a population sympatric with the predator (*Garcia et al., 2012*). Taken together, and whether from the perspective of the native or non-native amphibian, a species' ability to recognize and react to chemical cues from taxa which they do not share a recent evolutionary history with can have major impacts on their success. This is particularly true when the dynamics of species interactions change, such as the recent arrival or expansion of a non-native species takes place. African clawed frogs (*Xenopus laevis*) are a feral amphibian that has potential to have large impacts on native amphibians and as their invasive range increases, this warrants research to investigate native species' behavioral response—as well as their own.

*X. laevis* are native to Southern Africa (*Van Sittert & Measey, 2016*), but have been introduced to many countries around the world (*Measey et al., 2012*). *X. laevis* prey voraciously on a diversity of invertebrate and vertebrate animals in freshwater ecosystems (*Fibla et al., 2020*; *Lillo, Faraone & Lo Valvo, 2011*) and are likely successful invaders due to their generalist diet (*Courant et al., 2017*) and fast maturation times (*Rödder et al., 2017*). As such, there is concern that invasive *X. laevis* may outcompete native species for shared prey items or directly consume and extirpate native species (*Rödder et al., 2017*). In the United States, *X. laevis* have become well established in Arizona, California, Florida, and Washington (*Ojala-Barbour et al., 2021*). *X. laevis* in Washington are particularly troublesome because they have spread across multiple cities and counties in the south Puget Sound area and the frogs seem to persist in ponds that freeze in winter (*Ojala-Barbour*

*et al., 2021*). Although *X. laevis* were first discovered in Washington in 2015, the threat of *X. laevis* to native species in Washington or the broader Pacific Northwest region remains largely unknown, as well as the degree of its spread beyond the three known regions where it currently occurs. Determining the threat to native aquatic species could help identify and refine management targets (*Ojala-Barbour et al., 2021*). However, current management tools for *X. laevis* in Washington are also sparse as prior eradication efforts using trapping and poisoning have failed (*Ojala-Barbour et al., 2021*). Thus, there is an urgent need to understand how much of a threat *X. laevis* pose to native species, particularly in this region, and what tools might be available to manage *X. laevis*.

To address the knowledge gaps in our understanding of the degree of threat *X. laevis* pose to native amphibians we used chemical behavioral analyses to explore the threat of and management options for this feral frog. First, we tested whether larvae of a native amphibian species, *R. aurora*, respond to chemical cues from feral *X. laevis* differently than to native amphibian predator chemical cues. This goal emerged from observations showing that ponds without *X. laevis* have diverse native amphibian communities whereas adjacent ponds with *X. laevis* are devoid of native amphibian species (Fig. 1; Friesen et al., unpubl). Second, we assessed whether native rough-skinned newts (*Taricha granulosa*) could be an effective biocontrol against *X. laevis* by testing whether feral *X. laevis* responded to newt chemical cues (including toxins). This goal emerged from two observations in early 2022. First, students at Saint Martin's University began assisting the migration of newts across fence barriers that were meant to stop *X. laevis* from spreading (Fig. 2). Although we were regularly catching, marking, and releasing *X. laevis* in the preceding fall, our trapping in Lacey, WA yielded no *X. laevis* once additional newts were added to the pond, despite concurrent trapping effort in Issaquah (similar latitude, ∼100 km east) that yielded hundreds of *X. laevis* in similar sized ponds over the same timeframe. Second, we temporarily housed an *X. laevis* with a *T. granulosa* in our husbandry facilities which resulted in the *X. laevis* dying in less than 24 h. These two observations led to the hypothesis that *X. laevis* avoided and/ or were harmed by *T. granulosa* toxins or other cutaneous chemicals. *T. granulosa* is native to Western Washington and with other members of the genus *Taricha* have been the subject of intense study due to their robust cutaneous toxins, particularly tetrodotoxin (TTX; *Vaelli et al., 2020*). Research has shown that aqueous toxins exuded from these newts can elicit an antipredator behavioral response in larval amphibians, reduce the predatory success of dragonfly larvae, and cause invasive snails to migrate away (*Zimmer et al., 2006*; *Bucciarelli & Kats, 2015*; *Ota et al., 2018*). Accordingly, we predicted that native amphibian larvae would elicit an anti-predator response to a native newt but not an *X. laevis* and that *X. laevis* would be deterred by *T. granulosa* chemical cues.

## METHODS

### Species and sites

We studied feral *X. laevis* captured from stormwater ponds in Lacey and Issaquah, WA and housed in a captive facility at the Saint Martin's University campus in Lacey. This
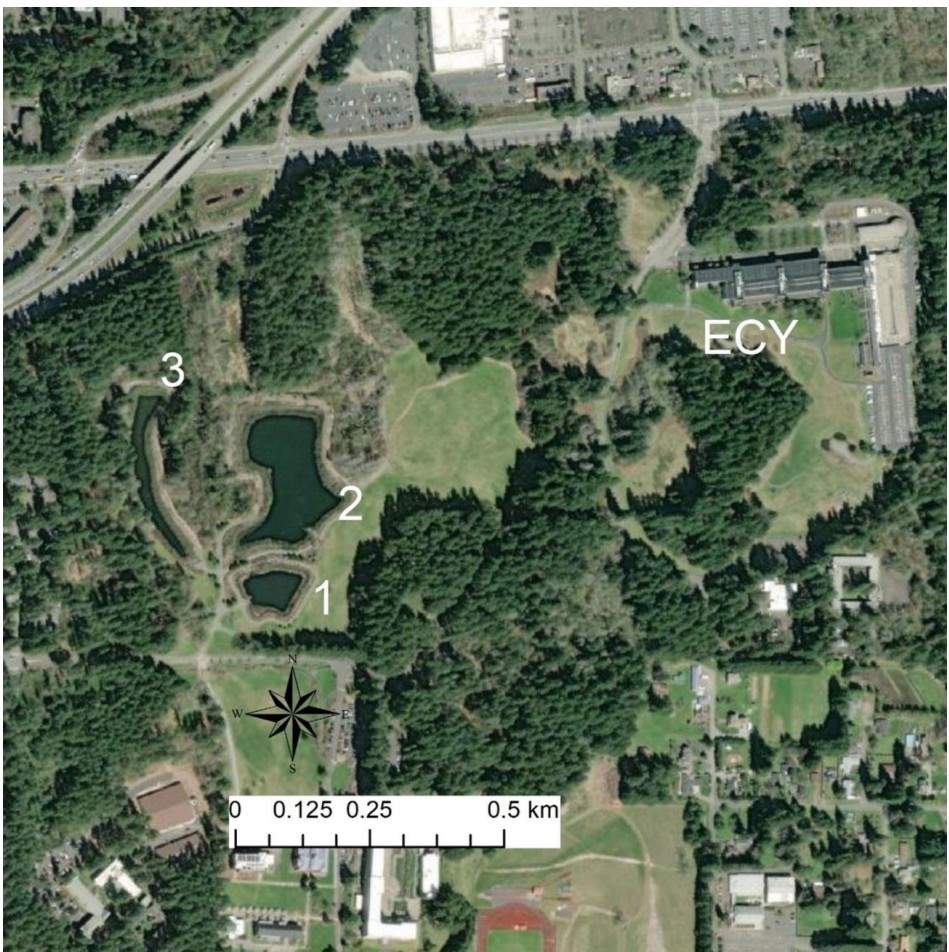

**Figure 1 Map of sites.** Map of focal source ponds. ACF are present in high densities in Pond 1, are rare in Pond 2, largely absent from Pond 3, and have not been detected in the ECY pond. Trapping has found chorus frog (*Pseudacris regilla*) and long-toed salamander larvae (*Ambystoma macrodactylum*) in Pond 1 and 3. Red-legged frog (*Rana aurora*)—a relatively urban-sensitive species—breed in Pond 3 in addition to the other native amphibians from Pond 1 and 2, and the ECY pond. Our experimental ACF were sourced from Pond 1, *R. aurora* embryos from ECY pond, and newts were collected between Ponds 1 and 2.

research was done under Saint Martin's University animal ethics permit SMUAE 22_1. State permissions were under the programmatic permit issued to WDFW employees for capturing and handling wildlife. Native species were captured from stormwater ponds (Ponds 1, 2, and 3) also in Lacey, WA, where *X. laevis* are not present (Fig. 1), with permissions from Washington Department of Fish and Wildlife. On 24 March 2022, two partial *R. aurora* egg masses (∼50 embryos each) were collected from the Ecology (ECY) stormwater pond ∼1 km northeast of the Lacey stormwater ponds where *X. laevis* inhabit (Fig. 1). *X. laevis* are not known to inhabit the ECY pond. *X. laevis* were collected from Pond 1, and newts collected between Pond 1 and 2. *T. granulosa* must reproduce in water

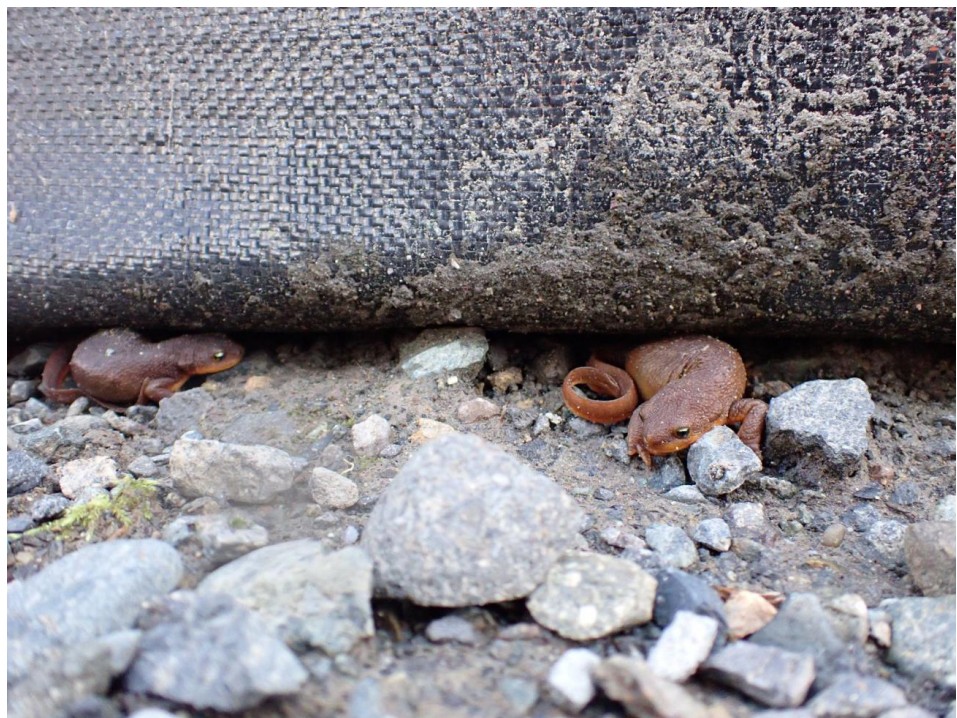

**Figure 2** **Example of newts trapped along fence.** Native rough-skinned newts migrating to breed at Ponds 1 and 2 but became stuck along silt fencing meant to limit ACF dispersal from these ponds. In 2022, St. Martin's students assisted 1,207 newts over the fences so they could breed. Imagery courtesy of Esri (Redlands, CA, USA).

and can either live permanently in water or migrate upland after breeding. These newts are predators of amphibian larvae and so *X. laevis* may compete with newts for food.

*X. laevis* and five native newts were housed in small groups in 38 L tanks and fed dehydrated and frozen blood worms during the duration of our trials, with tanks cleaned daily or every other day. Tadpoles were housed independently in 0.47 L plastic containers and fed ground up fish flakes (Omega One Super Color Flakes) every other day. Animals were housed in the lab (not euthanized) after trials for future research. *T. granulosa* were captive for at least two weeks prior to any trials.

## Predator cues

We housed both partial egg masses together and *R. aurora* embryos hatched in aged tap water at room temperature from 25 May–22 June 2022. We exposed tadpoles (Gosner stages 24–42) to chemical cues from *T. granulosa* (a native newt predator) and feral *X. laevis.* The tadpoles developed during the trials. *R. aurora* tadpoles were collected from an adjacent pond where no *X. laevis* were present. We made a chemical stimulus solution by soaking an adult newt or *X. laevis* in 300 mL of aged tap water for two hours in separate 0 .47 L containers (Fig. 3). Untreated aged tap water was used as a control. After 2 h, the adult amphibians were returned to the housing enclosures. We pipetted two mL of the *X. laevis* cue, newt cue, or control water into *R. aurora* tadpole experimental containers containing

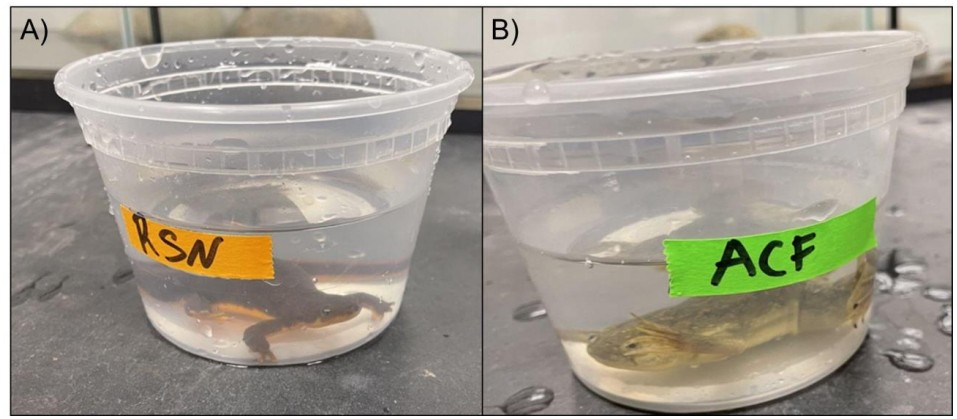

**Figure 3** (A–B) Experimental set up. Chemical cues were extracted from native rough-skinned newts and feral ACF by bathing them in 300 mL of aged tap water for 2 h. Larval *R. aurora* were then exposed to two mL aliquots of either chemical cue or control water with no amphibian cues.

200 mL of aged tap water. The tadpoles were allowed 2 min of acclimation prior to recording behaviors. After the acclimation period, we recorded tadpole behaviors for 10 min. At least three trials of each treatment were conducted each day. We completed 90 trials (28 *X. laevis* cues and 31 each for newt cues and controls) using a total of 17 *R. aurora* tadpoles. Over the duration of our study, we exposed most tadpoles to all three treatments (control and two cue treatments), although some tadpoles were only exposed to two different treatment types across the study due to logistical constraints. Three replicates of each treatment were done each day and tadpoles were assigned to treatments to ensure they were exposed to different treatments in subsequent trials. Experiments occurred at room temperature and no refugia were added given the small size of the experimental containers. We scored *R. aurora* larval behaviors into four behavior categories and recorded duration of each: Nothing, Foraging, Swimming, and Frantic Swimming. We defined "Nothing" as sedentary tadpoles displaying no movement, "Foraging" as tadpoles exhibiting mouth movements and pecking at the bottom of the experimental containers, "Swimming" as constant, slow movements in circular patterns around the containers, and "Frantic Swimming" as rapid, erratic movements in variable directions.

## Native newt biocontrol

Between 9 June 2022 and 8 September 2022, we performed behavioral choice tests on *X. laevis* exposed to *T. granulosa* to test whether *X. laevis* responded to newt cues. Adult *X. laevis* and newts were used in the biocontrol experiment and each animal was randomly selected from our husbandry facility. Each choice test was conducted in 2 L of aged tap water inside of a rectangular 38 L aquarium. The aquarium was divided into five sections along the long axis (Fig. 4). Mesh pouches made of black window screening were placed inside of the aquarium, adjacent and parallel to each of the two short sides (Fig. 4). One pouch was empty (control) and the other contained a newt (treatment). At the initiation of the experiment, we manually agitated each newt for 1 min, by gently stroking the

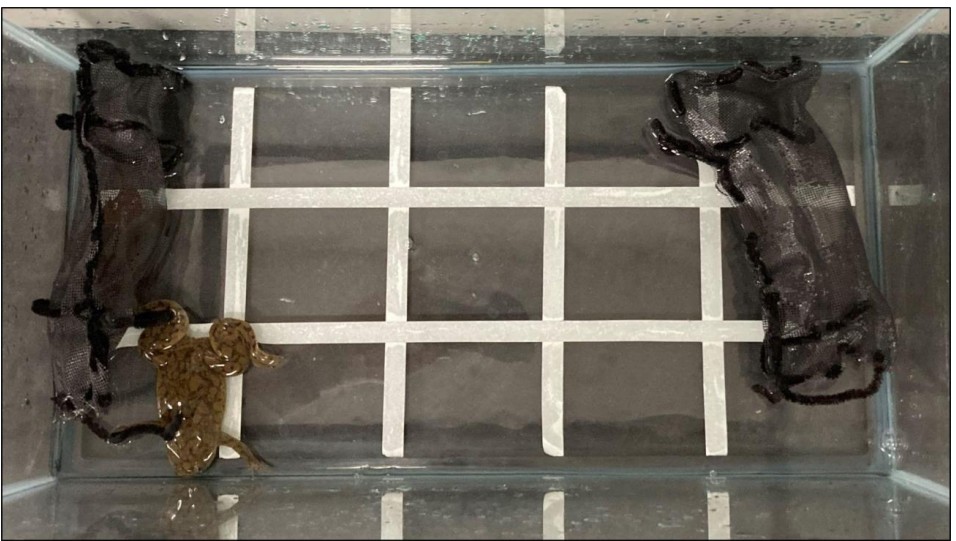

**Figure 4 Experimental set up.** Experimental set up to test whether feral ACF responded to native rough-skinned newt cues. ACF were placed in experimental tanks with two mesh bags: one containing a newt and an empty control bag.

anterior and posterior sides to promote the production of tetrodotoxin (*Bucciarelli & Kats, 2015*). Newt movement and direct interaction were 180° constrained by the use of sealed pouches but still allowed *X. laevis* to be exposed to chemical and visual stimuli. An *X. laevis* was placed in the center of the tank, parallel to the mesh pouches and facing out of the aquarium. For 10 min post-release, we observed *X. laevis* behavior and the duration spent at various positions within the enclosure. We recorded *X. laevis* positions based on where they occurred across the five sections in the enclosure and the total amount of time spent in each section. When the *X. laevis* was on the section with the newt or the section adjacent to the newt, its position was recorded as "Newt" (Fig. 4). When the *X. laevis* were in the middle fifth section, the time was recorded as "Center". When the *X. laevis* was on the section with the empty mesh bag or the section adjacent to the empty mesh bag, the *X. laevis*'s position was recorded as "Away" from the newt. We performed a total of 50 *X. laevis* behavioral choice tests: 25 with the newt on the southwest side of the aquarium and 25 with the newt on the northeast side of the aquarium. We switched which side of the aquarium that newts were placed to ensure *X. laevis* were not responding to other confounding cues in the laboratory.

## TTX analysis

We collected toxin samples from *T. granulosa* used in trials by repeatedly stroking the dorsal region of a newt anterior to posterior for one minute and then soaking it in 100 mL of aged tap water for one hour. After soaking the water solution was aliquoted into 1.5 mL screw cap microtubules. The samples were prepared for TTX analysis following methods outlined in *Ota et al. (2018)*. All samples were analyzed using a Shimadzu high-performance liquid chromatography system with fluorescence detection (*Bucciarelli et al., 2014*). The

detection limit of the system is below femtomolar concentrations. We evaluated peak area of chromatograms against known TTX standards to determine whether TTX was present in solutions and if so, the approximate molar concentrations.

### Statistical analyses

*Predator cues:* For the predator cue data, we used linear mixed effects model (lmer function, 'lme4' package; *Bates et al., 2015*) and likelihood ratio tests (anova function) to test whether *R. aurora* tadpole behavior differed between the three treatments (Newt or *X. laevis* cues and Controls). We performed models for each of the three active behaviors separately (excluding 'Nothing'). For random effects, we used tadpole identity as well as day-of-year (DOY) as a proxy for tadpole ontogeny and because tadpoles were used for the same treatment type on different days. We visually checked model fit.

*Native newt biocontrol*: We used linear mixed effects models (*lmer* function, 'lme4' package; *Bates et al., 2015*) and likelihood ratio tests (*anova* function) to test whether feral *X. laevis* spent disproportionately more time near or away toxic native newts. We used trial day as a random effect in these models and our global model included the two fixed effects of Choice and Side. "Choice" included the three categories—Newt, Center, or Away—which represent the three regions of the experimental tanks where *X. laevis* spent time. The "Center" category was indicative of a frozen behavior, while movement towards the newt was considered "Newt", and movement opposite was classified as "Away". The "Side" category reflected the northeast or southwest orientation of the experimental tanks where newts were placed on each side for half of the trials. We used likelihood ratio tests to compare the global model to two reduced models containing only one variable and to compare the univariate models to a null model. If Choice was significant, we used Tukey's post hoc tests (*glht* function, 'multcomp' package; *Hothorn, Bretz & Westfall, 2008*) to assess pairwise differences among Newt, Center, or Away choices. We performed all statistical analyses in R version 4.0.4 (*R Core Team, 2020*).

## RESULTS

*Predator cues*: Our models on individual behaviors found differences in *R. aurora* behavior ($p = 0.03$). Tukey's post-hoc tests found that *R. aurora* tadpole Swimming rates were reduced in the newt treatment compared to Control treatments ($p = 0.05$). Tadpole rates in response to *X. laevis* were statistically indistinguishable from both the Control ($p = 0.87$) and Newt treatments ($p = 0.15$). For Frantic Swimming ($p = 0.09$) and Foraging ($p = 0.89$), our models found no differences in *R. aurora* behavior among treatments.

*Native biocontrol*: Linear mixed effects models and likelihood ratio tests supported a model containing only the variable Choice (Fig. 5; $p = 7.99$ e -14). Tukey's post hoc tests found that all pairwise comparisons were significant (Center *vs* Away $p = 1.0$ e $-0.4$, Newt *vs* Away $p = 0.003$, Newt *vs* Center $p = 1.0$ e -04) such that feral *X. laevis* spent the least time in the Center third of the tanks (mean = 8.9 s, $\pm$ 1.0 s SE), intermediate amounts of time Away from newts (mean = 210.6 s, $\pm$ 4.9 s SE), and the most time next to the newts (mean = 368.5 s, $\pm$ 5.1 s SE).

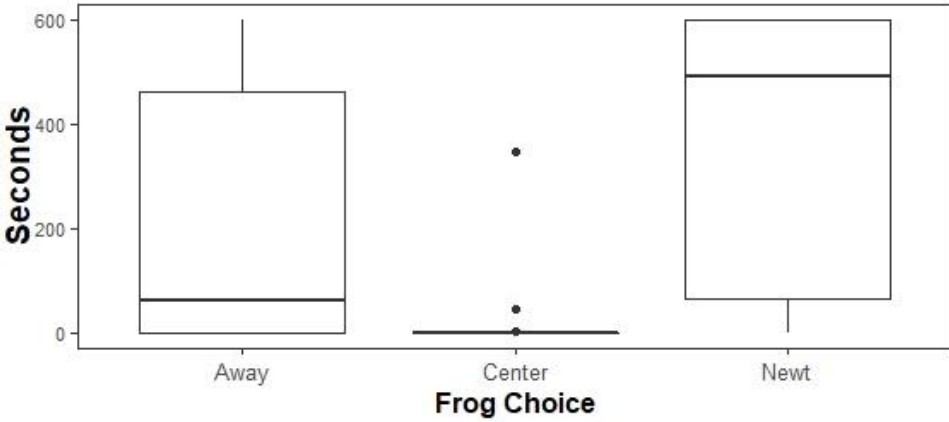

**Figure 5** **Time allocation of ACF to treatments.** Time spent by ACF away from native, toxic newts in sham controls (Away), next to native newts (Newt), or in the center of experimental tanks. ACF spent the most time next to newts, intermediate amounts of time next to sham controls, and the least amount of time at the center. Boxplots center lines indicate data medians, box edges represent 25th and 75th quartiles, and points are outliers.

*TTX Analysis*: We did not detect TTX in the sample solutions. Chromatograms showed no peak at the standard-derived elution time for TTX (Figure A1). It is possible that there are TTX analogues in the sample, but without commercially available standards, identification in the scope of this study is not possible. In general, the lack of a TTX peaks in the chromatogram indicates that TTX was at concentrations lower than 10x-15 moles/liter or possibly not present.

## DISCUSSION

Our study adds to the growing body of evidence that feral *X. laevis* pose a threat to native aquatic species (*Kruger et al., 2019*; *Lafferty & Page, 1997*; *Lillo, Faraone & Lo Valvo, 2011*). Feral *X. laevis* may be a concerning, hard-to-manage invasive predator in the Pacific Northwest. Our results show that a native species may not recognize *X. laevis* as a predator and that toxic *T. granulosa* may be challenging to use as native biocontrols against *X. laevis*; at least in the short-term. In our experiments, native *R. aurora* tadpoles exhibited strong anti-predator responses to native newt chemical cues by decreasing Foraging and increasing Frantic Swimming, but did not respond to *X. laevis* chemical cues. Interestingly, despite native tadpoles responding strongly to newt chemical cues, feral *X. laevis* did not respond to newts. These results underscore the threats that *X. laevis* poses to native species as a predator with few effective management options (*Ojala-Barbour et al., 2021*).

*R. aurora* tadpoles exhibit more antipredator behavior towards native newts than to feral *X. laevis.* Newts elicited a classic anti-predator behavioral syndrome in tadpoles by causing tadpoles to be sedentary with bouts of Frantic Swimming compared to more typical cruising Swimming and Foraging behaviors (*Watkins, 1996*; *Laurila, Kujasalo & Ranta, 1997*; *Van Buskirk & Mccollum, 2000*; *Bridges, 2002*; *Gabor et al., 2019*). *X. laevis* cues elicited no such response in *R. aurora* tadpoles. These findings suggest that native

Pacific Northwest amphibians have evolved to exhibit antipredator behavior towards native predators but are unable to recognize invasive amphibian predator cues. This indicates *R. aurora* tadpoles are potentially vulnerable to *X. laevis* predation. However, the overall predation risk from *X. laevis* to *R. aurora* remains unclear as we did not perform feeding trials. Further research could clarify whether invasive *X. laevis* consume native amphibian larvae at high enough rates to cause population-level impacts. Additionally, because continued exposure to a predator cue can change the response of the cue receiver, it is possible that responses could have changed over the course of the trials (*Kruger et al., 2019*).

For this work, we focused on antipredator behaviors in *R. aurora* tadpoles—a species that is regionally listed as Stable, despite experiencing population declines primarily due to forest loss (*Washington Herp Atlas, 2009*), but has been listed as imperial is other part of its range (*e.g.*, within Canada; *Environment Canada, 2016*). Beyond the direct potential impacts to *R. aurora*, like predation, this work highlights how *X. laevis* may be a threat to other native species—including more sensitive species—which may not recognize it as a predator. For instance, our source *X. laevis* population in Lacey, WA is less than 35 km away from known populations of federally threatened *R. pretiosa* in Thurston County. Given the close proximity, of invasive *X. laevis* to federally listed amphibians, there is a need to proactively manage the spread of *X. laevis* and understand impacts to sensitive species, particularly if these species are naive to *X. laevis* predator cues. Beyond impacts to amphibians, there is also a need to understand potential impacts to native fishes. For instance, the same nearby habitats that host federally threatened *R. pretiosa* also are home to olympic mudminnows (*Novumbra hubbsi*), a state-sensitive species that is small (<80 mm long) and potentially vulnerable to *X. laevis* predation. Furthermore, diverse salmonid species occur near invasive *X. laevis* populations in Washington (*Ojala-Barbour et al., 2021*) that may also fall with this invasive frog's dietary range. Salmonids in the Pacific Northwest are culturally, ecologically, and economically important and several are listed under the U.S. Endangered Species Act (*Quinn, 2004*). Invasive *X. laevis* have been repeatedly detected in and adjacent to water bodies with various salmon species, including kokanee salmon (*Oncorhynchus nerka*). Although adult salmon are too large for *X. laevis* to consume, embryonic and fry life stages may be vulnerable to *X. laevis* predation, particularly if salmon are naive to *X. laevis* predator cues.

We anticipated that native newts might serve as a potential biocontrol agent against *X. laevis*. Although the neurotoxin TTX has been extensively studied in *Taricha* newts for its anti-predatory properties (*Zimmer et al., 2006*; *Bucciarelli & Kats, 2015*; *Ota et al., 2018*), to our knowledge it has not been studied for potential biocontrol purposes. We were motivated to test whether *T. granulosa* might be an effective biocontrol because several casual observations suggested that *X. laevis* may be sensitive to newt toxins. In particular, we anticipated that *T. granulosa* would be so toxic as to elicit a relatively rapid behavioral response in *X. laevis*. However, the presence of newts in our study appeared to attract rather than deter *X. laevis*. There are multiple reasons for this. First, we conducted relatively short-duration trials to assess *X. laevis* behavior. Longer trials may reveal different patterns if aqueous TTX takes longer than 10 min to influence *X. laevis* physiology. Second,

additional work may benefit from testing different densities of newts as higher doses of TTX may be needed to influence *X. laevis*. Third, our experiments did not allow *X. laevis* to directly interact with newts. Although we attempted to stimulate TTX in the *T. granulosa*, our experimental design limited interspecific interactions that could have produced ecologically relevant exposures. Regardless, the potential utility of *T. granulosa* as a biocontrol is probably greater through passive toxicity rather than through consumption. Other types of biocontrol could include large invertebrates, which *X. laevis* have been shown to exhibit antipredator behavior to (*e.g.*, measured as a decrease in activity when exposed to a predatory beetle, *Dytiscus dimidiatus*, and crayfish, *Procambarus clarkii*; *Kruger et al., 2019*). Finally, *X. laevis* may be attracted to visual cues more so than chemical ones. One study found that removing *X. laevis* was most successful when traps were baited with conspecifics (*Lorrain-Soligon et al., 2021*); this result, in tandem with our findings, suggests that *X. laevis* may generally respond to visual cues like movement. Future studies may benefit from testing the response of *X. laevis* strictly to chemical cues. Although the ability to produce a powerful neurotoxin makes *Taricha* newts a tantalizing potential candidate for *X. laevis* biocontrol, additional research is needed to assess if this is a viable and ecologically neutral management option.

While our research indicated that invasive *X. laevis* are chemically cryptic predators that could pose a risk to native species and which are not readily deterred by newt chemical cues (including toxins), the chemical mechanisms underlying the relationships we explored warrants further attention. The newts used in our research were collected at our field site and kept in a laboratory setting for 1–2 months prior to our experiments. Because of the conflicting observations that motivated our experiment and our experimental findings, we analyzed aqueous newt extracts to determine if TTX was present and estimated concentrations. This analysis found no detectable TTX in the solutions which may have affected chemical cues between the newts and *X. laevis* in this study. Even so, this analysis found possible TTX analogues and/or relevant metabolites. While some research has indicated that TTX may increase in captive newts (*Hanifin, Brodie & Brodie, 2002*), other research shows lower TTX levels in newts compared to wild individuals (*Gall et al., 2022*). There is also evidence that TTX is linked to the newt microbiome, potentially indicating our captive setting did not allow for proper microbe growth (*Vaelli et al., 2020*; *Gall et al., 2022*). Further, TTX concentrations vary and fluctuate within and among *T. granulosa* populations (*Bucciarelli et al., 2016*; *Reimche et al., 2020*) and so our population may inherently maintain low amounts of TTX or at the time of sampling possessed relatively low toxin concentrations. Interestingly, our results clearly show that native *R. aurora* tadpoles respond to newt chemical cues, regardless of whether TTX or some other possible analogue was the constituent molecule of the solution. It may have also suggested their ability to detect it in concentrations, while our methods could not. These findings highlight new opportunities for understanding the chemical ecology of newts and their interactions with other species.

## CONCLUSION

We aimed to identify the roles that chemical cues play in mediating the relationships between invasive *X. laevis* and native amphibian prey and toxic newts. We found that: (1) native *R. aurora* tadpoles show strong anti-predator responses to newts but do not recognize *X. laevis* as predators and (2) *X. laevis* were attracted rather than deterred by *T. granulosa* chemical cues in short-duration trials. The lack of anti-predator responses to invasive *X. laevis* may provide a foraging advantage over native amphibian predators and suggest *X. laevis* have potential to have detrimental effects on native species populations. It is also possible that introduced *X. laevis* do not have a response to the defenses of native species because they have not co-evolved with the mechanism. Our work has begun to uncover some of the mechanisms that may allow *X. laevis* to threaten native species and highlights new areas of research to improve management of this global invader.

## ACKNOWLEDGEMENTS

We are also grateful to Jacie Fabela and Quin Butler for their assistance in animal care during our trials. Additionally, Rebecca Lavier, Hannah Dotterweich, Panos Stratis helped with animal trapping. We are also grateful to our three anonymous reviewers for their thoughtful comments.

### Funding

This work was supported by Washington Department of Fish and Wildlife. The funders had no role in study design, data collection and analysis, decision to publish, or preparation of the manuscript.

### Grant Disclosures

The following grant information was disclosed by the authors:
Washington Department of Fish and Wildlife.

### Competing Interests

Max Lambert is an Academic Editor for PeerJ.

### Author Contributions

- David Anderson conceived and designed the experiments, performed the experiments, analyzed the data, prepared figures and/or tables, authored or reviewed drafts of the article, and approved the final draft.
- Olivia Cervantez conceived and designed the experiments, performed the experiments, analyzed the data, prepared figures and/or tables, authored or reviewed drafts of the article, and approved the final draft.
- Gary M. Bucciarelli performed the experiments, analyzed the data, prepared figures and/or tables, authored or reviewed drafts of the article, and approved the final draft.

- Max R. Lambert conceived and designed the experiments, analyzed the data, prepared figures and/or tables, authored or reviewed drafts of the article, and approved the final draft.
- Megan R. Friesen conceived and designed the experiments, analyzed the data, authored or reviewed drafts of the article, and approved the final draft.

## Animal Ethics

The following information was supplied relating to ethical approvals (*i.e.*, approving body and any reference numbers):

Saint Martin's University animal ethics permit SMUAE 22_1.

## Field Study Permissions

The following information was supplied relating to field study approvals (*i.e.*, approving body and any reference numbers):

Washington Department of Fish and Wildlife.

## Data Availability

The raw behavioral data is available in the Supplemental Files.

## Supplemental Information

Supplemental information for this article can be found online at http://dx.doi.org/10.7717/peerj.17307#supplemental-information.

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
