# Peer review of "Feral frogs, native newts, and chemical cues: identifying threats from and management opportunities for invasive African Clawed Frogs in Washington state"

_PeerJ, doi:10.7717/peerj.17307_

## Round 0.1 · original submission · Major Revisions

This is an interesting study, and one that will be of keen interest for those working in both the applied and theoretical sides of invasion science, as well as in the field of the conservation of native herpetofauna. The reviewers both saw the value and merit of this work, and I agree. However, there are some key issues that will need to be addressed, as well as the normal flow of helpful comments and suggestions for manuscript improvements. Notably Reviewer 1 raise several concerns with respect to study/statistical design of some of your tests, some of which may be covered by a deeper explanation, but other sections that I agree may need to be re-analysed. They do provide some guidance as to how to tackle this. Both reviewers note that there is also a need to incorporate more background information and supporting citations to shore up some of your arguments and statements, and have provided a number of great references to get you starts in this respect as well.

All in all, the reviewers have some great guidance on how to strengthen the presentation of your work, and I hope you take care to consider their suggestions. And with respect to experimental design and data analysis, I understand having to go back and re-do stats is a pain. But our goal here is to find a way that allows you to present your work in the clear and accurate manner, that is as defensible as possible. So please do take the time to give considerable thought the comments regarding issues and limitations of your study and how the data is analysed. This may require restructuring data collections (e.g., dividing and removing the center zone) or reanalysing data (e.g., after removing or changing a variable) or keeping an aspect of the work that may be somewhat confounded/muddled by study design, but making sure to explain it’s limitations appropriately. Overall, try to look at each comment as a means to make your paper better and stronger, and revise the manuscript and study with that in mind.

Once you have been able to address the comments and make your revisions, I am looking forward to reading the next iteration of this work.

Reviewer 1 ·

Basic reporting

See attachment

Experimental design

See attachment

Validity of the findings

See attachment

Additional comments

none

Annotated reviews are not available for download in order to protect the identity of reviewers who chose to remain anonymous.

Reviewer 2 ·

Basic reporting

Title: Feral frogs, native newts, and chemical cues: identifying threats from and management opportunities for invasive African Clawed Frogs in Washington state

The manuscript provides relevant information about an invasive anuran species in the Pacific Northwest. The topic discussed in the paper is of great importance and appropriate to this journal. The authors presented valuable data collected in experiments and discussed potential management strategies for African clawed frogs. I consider critical to highlight how Xenopus laevis can be affecting native amphibians and other organisms as reported previously in the literature for places in Europe and also for places in the PNW.
The authors need to include references of previous studies done in the invaded area to improve their paper (introduction and discussion). Description of the experiments needs improvement as some details are missing.

Basic Reporting
The article is clear but its intro and background can include more references to the topic to be improved. Relevant references about Xenopus laevis are missing and including those will support and strength the content of this paper.

Some references authors can cite:

• Wilson, E.A., Dudley, T.L. & Briggs, C.J. Shared behavioral responses and predation risk of anuran larvae and adults exposed to a novel predator. Biol Invasions 20, 475–485 (2018). https://doi.org/10.1007/s10530-017-1550-x
• McCoid MJ, Fritts TH (1980) Notes on the diet of a feral population of Xenopus laevis (Pipidae) in California. Southwest Nat 25:272–275
• Lillo F, Faraone FP, Valvo Lo M (2011) Can the introduction of Xenopus laevis affect native amphibian populations? Reduction of reproductive occurrence in presence of the invasive species. Biol Invasions 13:1533–1541
• Wilson et al (2018) Invasive African clawed frogs in California: A reservoir for or predator against the chytrid fungus. PLOS ONE https://doi.org/10.1371/journal.pone.0191537


Movement
• Measey GJ, Tisley RC (1998) Feral Xenopus laevis in South Wales. Herpetol J 8:23–27
• Eggert C, Foquet A (2006) A preliminary biotelemetric study of a feral invasive Xenopus laevis population in France. Alytes 23:144–149
• Faraone FP, Lillo F, Giacalone G, Lo Valvo M (2008) The large invasive population of Xenopus laevis in Sicily (Italy). Amphibi-Reptil 29:405–412

The authors can improve the map of focal source ponds (Fig 1) including an inset (reference map) , moving the north star to one of the corners using a white background so it can be easily seen and changing the scale. Although ACF is defined in the text, I suggest to use here Xenopus laevis instead of the letters. Caption can be summarized maybe the use of a map legend can help at least with description of densities.

Experimental design

The submission represents original research but the objectives of the research can be redacted in a different way to get a clear idea. This research fills a knowledge gap and it seems to be performed with high technical standards. Details in the methods will improve the paper making their experiments reproducible by different researchers.

Validity of the findings

This manuscript brings attention to a topic that needs to be addressed with urgency. Authors generated information about a potential strategy for management via experiments and previous observations. I considered the manuscript as appropriate to be published after revisions as there is a need to include references for the introduction and discussion, and details in the methods that will improve the quality of the paper. Conclusions are linked to objectives and related to results.

Annotated reviews are not available for download in order to protect the identity of reviewers who chose to remain anonymous.

---

## Round 0.2 · Major Revisions

I am please to inform you that we have received a second round of reviews for your manuscript, which will no doubt continue to improve and polish this write-up into what I feel will become a wonderful paper.

During the second round of revisions, Reviewer 2 was unable to partake, but given some of the concerns raised by Reviewer 1, we felt having another set of eyes (and ideas) would be the best course of action. This is why you will find comments and suggestions here from a third reviewer.

As we’d expect both Reviewers 1 and 3 share some similar suggestions for areas that could be improved, such as continuing to bolster the amount of background information on the system and relevant associated citations. Reviewer 3 provides quite a few citations for studies that you may find useful, in this respect.

Furthermore, both have raised some concerns about the analytical methods. From Reviewer 1, some related back to their questioning of the value of conducting a PCA over simply looking at the behaviours independently. In your rebuttal you explain that the purpose of your approach is to collapse correlated variables to create a new proxy variable; in doing so keeping the analysis tidy. I can see value in the idea that combined could be seen as tidier, but I agree with Reviewer 1 that in this case it is actually doing the opposite. I also agree with Reviewer 1 that you should remove the “no movement” behaviour, as it is just the opposite of the other three. The tidier and more interpretable option is to run three linear models on foraging, slow swimming, and frantic swimming. In doing so I think the findings are more directly informative. And would make for stats that would provide you far more support to make statements like “… that larvae exposed to Newt cues displayed more frantic swimming and less foraging than Control larvae or larvae exposed to ACF.”. It would also be simply easier for the reader to clearly see and interpret and in reality, it is not more complex or untidy than first running a PCA, reporting the loading, selecting a PC and then running a linear model. In fact, I feel that this process appears far less tidy, is harder for readers to directly understand, and limits how you can talk about the different behaviours independently (as it is not was you are directly testing with a PCA).

I know that having a review come back that asks you to redo your stats is annoying, but the goal here is to present your work in a way that is correct, clear, and interpretable – PCAs have their value, but I am not seeing it here. The extra steps of a PCA I feel muddy the water of what could be simply three straightforward linear models (models that could account for all the additional variables and controls, like Reviewer 3’s suggestion of adding in Gosner stage).

Reviewer 3’s, fresh set of eyes on this also picked up a number of areas where some greater clarity could be included (see their comments). Ultimately, some of the information may be within the manuscript in other areas, but if the writing is bringing in some ambiguity for a reviewer, no doubt it will also cause issues with readers; which of course we want to avoid. Furthermore, Reviewer 3 makes a good many other excellent suggestion for additions and improvement that can be implemented here.

As I said, I absolutely understand that a second round of reviews that call for re-analysis of the data is not what authors typically want to hear. But I do think that straightforward analyses are typically more informative, compared to those made unnecessarily more complex. For a reader to trust and understand statement referring to tadpoles that frantic swim more and forage less when expose to newt cue, compared to control or ACF cues, it is very easy with a model that looked at foraging and one that looked at frantic swimming and include a p value for each. It is more complicated and complex when the reader has to sift through PCA score and weighting and then see which one is which, how does PC1 differ from PC2, 3, for 4 – and what are the weightings for them all. If you truly feel that this is a far more rigorous and informative approach then I would be more than happy to hear your arguments for it – but if the argument for a PCA over direct linear models is tidiness (e.g., more clear and concise), then I am not sure about that. I would, very much, suggest switching to using three direct models that look at your each of the 3 behaviours (forage, swim, frantic) for your newt, ACF, and control.

I know that with some more refining, editing, and a round of reanalysis this study's write-up will only keep getting better and better. I am looking to forward in reading your next version.

Reviewer 1 ·

Basic reporting

See attached file

Field background partially covered

Experimental design

See attached file

Missing information in some places

Validity of the findings

See attached file

Some points raised about the statistical analysis

Discussion fail to address the various mechanisms for the lack of response, which itself is of interest.

Annotated reviews are not available for download in order to protect the identity of reviewers who chose to remain anonymous.

·

Basic reporting

The basic reporting of this paper is clear and the level of English throughout is acceptable. Although sufficient background and context is provided for the impact of invasive African Clawed Frogs, there is still a lack of background regarding the native species Rana aurora (Please see comments under ‘Introduction’ for more detail). The authors seem to discount all literature regarding ACF that is not relevant to the specific region, however, these are important to explore and can provide important context. This is needed to demonstrate how this work fits into the broader field of knowledge. The layout of the report is appropriate, however, figure 2 have been suggested to be removed and authors have denied this request. The manuscript is ‘self-contained’ meaning that the results are relevant to the hypotheses.

Experimental design

The experimental design displays original primary research. The research question is well-defined, and it has been stated how the research fills the knowledge gap. The research has been conducted in conformity with the prevailing ethical standards in the field. However, the methods are not clearly described with sufficient information to be reproducible by another investigator – more detail is required (see Methods for detailed comments).

Validity of the findings

Validity of the findings: Although the statistical analysis has been questioned by previous reviewers and I have made some suggestions myself (see methods), the data on which the conclusions are made are robust and controlled (see results for more specific comments). The conclusions are generally well stated (watch out for overselling findings) and linked to the original research question with some minor errors and omissions (see Discussion for more detailed comments).

Additional comments

Introduction:
After reviewing the initial comments from reviewer 1 and 2, I believe that some concerns are still not addressed in the introduction. Some of which I concur with. Both reviewers suggested adding more literature and context to the rationale of your study. Although your introduction is quite long it does not completely explore the literature enough to support your research question and aims. You do not mention any studies that discuss how ACF respond to threatening chemical cues, this will inform your predictions of experiment 2 (see Kruger et al 2019). Why did you choose to test adult ACF and not tadpoles? Some other information that is missing from the introduction is the native species (Rana aurora) information for experiment 1, what type of antipredator response do you expect to see from their interaction with native predators and how does it differ from ACF or other invasive species (see Kiesecker et al 1999, Kiesecker et al 1997 and Kiesecker et al, 2002)? Would a lack of response indicate that ACF have impact on R. aurora? I also understand that literature on the impact of ACF is limited for that region, however, ACF impact has been widely studied and should be considered here.

Here is some literature I suggest you read through and consider for your introduction:

Kiesecker, J.M., Chivers, D.P., Marco, A., Quilchano, C., Anderson, M.T. and Blaustein, A.R., 1999. Identification of a disturbance signal in larval red-legged frogs, Rana aurora. Animal Behaviour, 57(6), pp.1295-1300.

Kiesecker, J.M. and Blaustein, A.R., 1997. Population differences in responses of red‐legged frogs (Rana aurora) to introduced bullfrogs. Ecology, 78(6), pp.1752-1760.

Kiesecker, J.M., Chivers, D.P., Anderson, M. and Blaustein, A.R., 2002. Effect of predator diet on life history shifts of red-legged frogs, Rana aurora. Journal of Chemical Ecology, 28, pp.1007-1015.

Wilson, E.A., Dudley, T.L. and Briggs, C.J., 2018. Shared behavioral responses and predation risk of anuran larvae and adults exposed to a novel predator. Biological invasions, 20, pp.475-485.

L106: Remove extra comma.

L112: Specify that it is urgent to understand in this area, because the threat or impact of ACF has been studied in other areas.

L124: Lacey? Which pond did you introduce the newts?

L130: You can write T. granulosa as you’ve mentioned the species before.

Figure 1: In the text you describe that ponds with ACF present (1-3) have a lower native amphibian diversity than ponds without ACF (ECY). However, in the figure caption you describe trapping chorus frogs and long toed salamanders in ponds 1 and 3? And rough-skinned newts from ponds 1 and 2? This does not clearly demonstrate the lower diversity in ponds with high density ACF.

In figure 1 you clarify that newts moved between ponds 1 and 2, in which direction? You also mention that both these ponds (1 and 2) have ACF. Which one decreased in ACF when the newts were introduced? Are you assuming that the pond with ACF that decreased in numbers had no newts before the introduction?

Figure 2: I agree with the previous reviewer that figure 2 can be removed, however, I note that the authors do not agree. I will leave it for the editor to decide.

Methods:
Although you have clarified that native species were collected from ponds without ACF, you don’t use consistent language here. Please write that you collected ACF from pond 1, R. aurora from ECY and newts between ponds 1 and 2 (that do have ACF).

The previous reviewer 2 asked for more information to be added here regarding sample size and housing information (housed together or separate) which was not fully addressed. I concur with this review and agree that this information is needed.

L147: Please clarify what native species and how many, what was your sample size for both native and invasive species?

L149: You mention the abbreviation ECY but only later (L151) mention that it is the Ecology pond.

L159: Again, clarify how many tadpoles you tested. How many replicates do you have?

L162: R. aurora in italics

L160: The Gosner stages you tested are quite varied and studies have shown that amphibians can have stage-specific responses to predator exposure (see Relyea 2003). You cannot change that now, however, this should be included in your discussion and the effect of this should be included in your stats for example, as a random effect in your model.

Relyea RA (2003) Predators come and predators go: the reversibility of predator-induced traits. Ecology 84: 1840–1848, https://doi.org/10.1890/0012-9658(2003)084[1840:pcapgt]2.0.co;2

L166: Were all tadpoles tested together? Or individually? What study informed this approach?

L168: This continuous exposure to predator cue can increase experience of tadpoles and can influence tadpole response to predator cue (see Kruger et al 2019).

Tadpoles can react differently to a cue when they experience it for a first time than when they experience it for a second time. Again, not something you can address now, but I think this should be included in your statistical analysis and discussion.

L168: What constitutes a trial?

L170: 17 tadpoles are quite a small sample size.

L171: The design of the treatment exposure is still confusing and this sentence does not clarify what you have done. Why most tadpoles and not all?

Native newt control: Did you have a choice experiment where both pouches had nothing in? To discriminate between a reaction to newt cue or to the bag?
Was the water changed between treatments? Were the bags washed between treatments?

L195: Redundant, you already mentioned that the aquarium was divided into five sections (L187) and that the mesh bags were placed on the short sides.

Statistics:
Predator cue: What was the response variable? PC1, PC2 or PC3?

L226: We?

Did you perform any top model selection?

As mentioned earlier, I would suggest adding tadpole stage and exposure (1st or 2nd) as random effects in the model.

Results:

L254: I would suggest starting with PC1 results then PC2 etc.

TTX analysis: interesting, can you elaborate what you expected to see as this information and background is missing from the introduction?

Discussion:

L186: Kruger et al is definitely not the only paper in the growing body of evidence of ACF impact.

L187: Start this paragraph with your main findings not a speculation – ‘may’.

L288: ‘native species’ – do you mean the 1 species that you tested?

L288: You cannot say that toxic newts are challenging to use as you did not assess the ease of use of toxic newts. Instead, you investigated the reaction of ACF to the presence of a toxic newt. The wording here is incorrect.

It is interesting that you tested the anti-predator response of native tadpoles but chose to test adult ACF. Tadpoles of ACF would be interesting to test here as they cannot escape the effect of toxic newts. Also, as they are smaller, they might be more vulnerable to the small doses of TTX.

L293: Again, your wording is misleading. You tested one form of management – biocontrol. If you are referring to other studies that also display other effective management options, then cite it.

L300: You are generalising to all amphibians, although you only tested one. I would be careful to do this as other studies have found that native amphibians do react to invasive ACF (see Wilson et al. 2018) Is R. aurora known to be a model amphibian?
Wilson, E.A., Dudley, T.L. and Briggs, C.J., 2018. Shared behavioral responses and predation risk of anuran larvae and adults exposed to a novel predator. Biological invasions, 20, pp.475-485.

L305: There are studies that show that X. laevis feed on tadpoles. You can cite these in support of them potentially feeding on R. aurora tadpoles. They do not seem to be species specific as they even feed on their own tadpoles.

L326: Add references to these studies you talk about.

L336: Here you can mention that tadpoles might have a different or stronger response than adult ACF.

---

## Round 0.3 · Major Revisions

Hello Dr Friesen,

After receiving your last email we would be happy re-review you and your co-authors' manuscript. You are correct that a large component of the reason for the rejection did pertain to the apparent interested with keeping the PCA aspect of the analysis, which although was improved with removal of the "no movement" behaviour, still did not offer a wholly effective way to look at the data you have (for more see below). That said, it was not the only reason. For example, the last rebuttal read as if many of the comments and suggestion made by the reviewers were not being appropriately considered (in fact some response felt fairly flippant, if not curt at times) and the responses lacked information such as what was changed and where new text was added (e.g., Line #). For example, when a reviewer asked for more clarity regarding the chromatogram analysis "I do not know what to think of the chromatogram analysis. Authors say “However, there were apparent peaks of possible TTX analogues in the sample…”. I see peaks in Fig 7 but I cannot find any legend there or in the manuscript. What are these three peaks? How do we know they are possible TTX analogues? This sentence is not enough. I wonder whether the toxin analysis should be included at all. Maybe in discussion, authors could state that the analysis has been attempted but that the toxin was not detected. There is not much to say about it even if I understand this is frustrating." Here an expert in the field who is confused by how the information is being presented. But the response in the rebuttal was "It is not frustrating. It is what it is; that’s science. Most studies identify new questions as a result. We moved the chromatogram to an appendix figure.". This is not the only time a helpful point to increase clairy or reframe how something is presented is totally overlooked. Another example would be when another reviewers asked: "Why did you choose to test adult ACF and not tadpoles?". This is a completely valid question and one where the answer would be a useful statement to include in the paper to give the reader insight into the reasoning for your study design. However the rebuttal response was "Every study is logistically constrained and we only had capacity to house and study one life stage; in this case we used adults as they are the dominant predators in this system." A little curt and no mention of including useful context in your manuscript. This theme within the rebuttals is persistent.

I would very much recommend going back through the two previous rounds of reviewer comments and suggestions and giving each one reasonable consideration. It is important to note that if a reviewer is confused or feels information is missing or lacking, or whatever, then there is a good chance that so too would anyone else reading the journal article if it was to be published. Our goal, as editors, reviewers, and authors should be to use the peer-review process to continue to polish and improve - even if the review comments may feel frustrating or pedantic or because maybe you already explained it elsewhere in the manuscript, or whatever - if a section is causing fiction, effort to smooth and polish it has merit and is worthwhile to do so.

With respect to the statistical analysis, removing the PCA is the best course of action I think. As I explained in my previous response, re-examining the data with clear and straightforward analyses may be the cleanest option for testing some of your metrics. This could be something like running three mixed-effect and/or linear models on foraging, slow swimming, and frantic swimming. You could even contextualise this with the behaviours these actions represent, foraging is foraging, but swimming/normal movement in novel test arenas is often associated with exploration, whereas your frantic swimming is a clear metric for anti-predator behaviours (as the paper presents). In doing so I think the findings are more directly informative (e.g., exposure to the cue increases antipredator behaviour, as seen by a significant increase in frantic swimming). You could then test for trade-offs with either exploration or feeding, but keeping it simple and interpretable is key.

So I think from an analysis perspective you can do some re-jigging on the stats and even extend this to the predictions (and rationale for behaviour-linked predictions) more explicitly.

I am quite looking forward to seeing the next version of this manuscript. I hope you can find these, and the previous suggestions and comments, helpful in continuing to improve and strengthen this manuscript.

· Appeal

Appeal

We appreciate your feedback on this manuscript. I wanted to touch base with you as it seems the decision to reject this manuscript is largely based on our retention of the PCA. After discussing with co-authors, we would be happy to remove the PCA if it means that this manuscript can continue on to the next stage. The two lead authors of this manuscript were undergraduate researchers who have both moved on to other career opportunities and have worked hard on completing the reviewer comments aside from this change.

Please let me know if you would reconsider this decision pending the removal of this analysis.


· · Academic Editor

Reject

Given the amount of changes made, the responses to the reviewers, and your desire to retain the statistical approach of using a PCA to test behaviour, we feel that we are not able to accept this manuscript in it current state.

The experiments your team ran are fascinating, and I genuinely believe there are correct, clear, and straightforward ways to test your data and present your findings. But justifying a PCA here by stating that as behaviors are non-independent and that changes in these non-independent grouped behaviors may not always be strongly evidence when assessed in isolation, is not quite right. There are many reasons a wealth of research has used a few behaviours and tests them independently.

Given the comment has been brought up twice now, and your team seems to want to stay the course, I feel we may be at an impasse here.

I hope our work in providing comments, recommendations, and suggestions can continue to be useful as you progress with this manuscript.

---

## Round 0.4 · Minor Revisions

I think the authors have done a good job at re-analysing their data in a way that allows a clearer understanding of the findings. Overall, I think the science of the manuscript and how it was looked at should meet the standards we are looking for here. The next course of action is to make sure the writing is. During my last round of reviewing, which is here, I note many typos and other editorial issues that I have endeavoured to flag for you. Some of the manuscript layout and structure should be tightened up as well (e.g., general species info in the Intro instead of methods). Another recurring issue is inconsistent terminology, like common names vs scientific names vs acronyms, or the capitalisation of variable terms ... that sort of thing. So there is quite a few things to get done to make sure this manuscript is a bit more polished before it can be accepted. I have included a PDF where I have, not encompassingly, flagged things that should be addressed. I can only attach a PDF to this online system, but I will send a Tracked Changes Word doc to the corresponding author to help expedite their revision and polishing process.

In short, some key things to make sure get addressed, are:

1) Fix the typos, extra spaces, or other bits and bobs that are flagged and make sure there are not any more

2) Address inconsistent terminology and formatting of terms

3) Strengthen, or add, topic sentences for all of the paragraphs of the Intro and Discussion, and add summary/segue sentences to the ends to better improve readability and flow

4) Making sure each paragraph is standalone and covers an entire concept (i.e., two smaller paragraphs discussing the same idea should be merged)

All in all, these are mostly just comments that work to make sure that the audience reading this paper can follow along as cleanly and easily as possible. And that as a manuscript this piece of writing is putting its best foot forward to do the study justice. I hope you find this helpful.

---

## Round 0.5 · Minor Revisions

The manuscript is looks much better. There are just a few more edits and tweaks to go. I caught some more typos, some fixes to naming consistency, a few issues with reference list, and some additional text that should help improve flow, including where some additional references are needed. Once again, I made the edits in TrackedChanges, so I will attach a PDF here - but will directly sent the corresponding author the Word file for expedited editing so this manuscript can process quickly.

---

## Round 0.6 · accepted · Accept

I would like to congratulate the authors on all of the hard work they have put in to improving and polishing their manuscript. I think the latest version works well to showcase this interesting study. They have addressed many of the previous reviewer comments and have tackled the editorial suggestions made by me. I am happy to recommend this manuscript be accepted.